# Evolution of Patterns of Care and Outcomes in the Real-Life Setting for Patients with Metastatic GIST Treated in Three French Expert Centers over Three Decades

**DOI:** 10.3390/cancers15174306

**Published:** 2023-08-28

**Authors:** Maud Toulmonde, Derek Dinart, Mehdi Brahmi, Benjamin Verret, Myriam Jean-Denis, Françoise Ducimetière, Gregoire Desolneux, Pierre Méeus, Jean Palussière, Xavier Buy, Amine Bouhamama, Pauline Gillon, Armelle Dufresne, Clémence Hénon, François Le Loarer, Marie Karanian, Carine Ngo, Simone Mathoulin-Pélissier, Carine Bellera, Axel Le Cesne, Jean Yves Blay, Antoine Italiano

**Affiliations:** 1Department of Medical Oncology, Institut Bergonié, 33076 Bordeaux, France; a.italiano@bordeaux.unicancer.fr; 2Department of Epidemiology and Clinical Research, Institut Bergonié, 33076 Bordeaux, France; d.dinart@bordeaux.unicancer.fr (D.D.); s.mathoulin@bordeaux.unicancer.fr (S.M.-P.); c.bellera@bordeaux.unicancer.fr (C.B.); 3Department of Medical Oncology, Centre Leon Berard, 69373 Lyon, France; mehdi.brahmi@lyon.unicancer.fr (M.B.); armelle.dufresne@lyon.unicancer.fr (A.D.); jean-yves.blay@lyon.unicancer.fr (J.Y.B.); 4Department of Medical Oncology, Gustave Roussy, 94800 Villejuif, France; benjamin.verret@gustaveroussy.fr (B.V.); clemence.henon@gustaveroussy.fr (C.H.); axel.lecesne@gustaveroussy.fr (A.L.C.); 5Department of Epidemiology and Clinical Research, Centre Leon Berard, 69373 Lyon, France; myriam.jean-denis@lyon.unicancer.fr (M.J.-D.); francoise.ducimetiere@lyon.unicancer.fr (F.D.); 6Department of Surgical Oncology, Institut Bergonié, 33076 Bordeaux, France; g.desolneux@bordeaux.unicancer.fr; 7Department of Surgical Oncology, Centre Leon Berard, 69373 Lyon, France; pierre.meeus@lyon.unicancer.fr; 8Department of Radiodiagnostic and Interventional Radiology, Institut Bergonié, 33076 Bordeaux, France; j.palussiere@bordeaux.unicancer.fr (J.P.); x.buy@bordeaux.unicancer.fr (X.B.); 9Department of Radiodiagnostic and Interventional Radiology, Centre Leon Berard, 69373 Lyon, France; amine.bouhamama@lyon.unicancer.fr; 10Department of Radiation Oncology, Institut Bergonié, 33076 Bordeaux, France; p.gillon@bordeaux.unicancer.fr; 11Department of Pathology, Institut Bergonié, 33076 Bordeaux, France; f.le-loarer@bordeaux.unicancer.fr; 12Department of Pathology, Centre Leon Berard, 69373 Lyon, France; marie.karanian@lyon.unicancer.fr; 13Department of Pathology, Gustave Roussy, 94800 Villejuif, France; carine.ngo@gustaveroussy.fr

**Keywords:** metastatic GIST, tyrosine kinase inhibitors, clinical trials, locoregional treatments, real-life data

## Abstract

**Simple Summary:**

Gastrointestinal stromal tumors (GIST) are rare digestive mesenchymal tumors, and their treatment strategies have been revolutionized in recent decades with the discovery of tyrosine kinase inhibitors (TKI). This study assesses evolution of real-life treatment strategies for patients with metastatic GIST treated in three French expert centers over 30 years, including access to clinical trials and locoregional procedures to metastasis, and their impact on survival. The main results are the high levels of patient inclusion in clinical trials and locoregional treatments of metastasis, that translate in a benefit in survival, together with the absence of significant differences in the magnitude of benefit from non molecularly driven use of various TKI in later lines since the introduction of imatinib. This study advocates for early referral of metastatic GIST patients to expert centers to orchestrate the best access to future innovative clinical trials together with locoregional strategies and further improve GIST patients’ survival.

**Abstract:**

Gastrointestinal stromal tumors (GIST) are rare mesenchymal tumors characterized by *KIT* or *PDGFRA* mutations. Over three decades, significant changes in drug discovery and loco-regional (LR) procedures have impacted treatment strategies. We assessed the evolution of treatment strategies for metastatic GIST patients treated in the three national coordinating centers of NetSarc, the French network of sarcoma referral centers endorsed by the National Institute of Cancers, from 1990 to 2018. The primary objective was to describe the clinical and biological profiles as well as the treatment modalities of patients with metastatic GIST in a real-life setting, including access to clinical trials and LR procedures in the metastatic setting. Secondary objectives were to assess (1) patients’ outcome in terms of time to next treatment (TNT) for each line of systemic treatment, (2) patients’ overall survival (OS), (3) evolution of patients’ treatment modalities and OS according to treatment access: <2002 (pre-imatinib approval), 2002–2006 (pre-sunitinib approval), 2006–2014 (pre-regorafenib approval), post 2014, and (4) the impact of clinical trials and LR procedures on TNT and OS in the metastatic setting. 1038 patients with a diagnosis of GIST made in one of the three participating centers between 1990 and 2018 were included in the national prospective database. Among them, 492 patients presented metastasis, either synchronous or metachronous. The median number of therapy lines in the metastatic setting was 3 (range 0–15). More than half of the patients (55%) participated in a clinical trial during the course of their metastatic disease and half (51%) underwent additional LR procedures on metastatic sites. The median OS in the metastatic setting was 83.4 months (95%CI [72.7; 97.9]). The median TNT was 26.7 months (95%CI [23.4; 32.3]) in first-line, 10.2 months (95%CI [8.6; 11.8]) in second line, 6.7 months (95%CI [5.3; 8.5]) in third line, and 5.5 months (95%CI [4.3; 6.7]) in fourth line, respectively. There was no statistical difference in OS in the metastatic setting between the four therapeutic periods (log rank, *p* = 0.18). In multivariate analysis, age, AFIP Miettinen classification, mutational status, surgery of the primary tumor, participation in a clinical trial in the first line and LR procedure to metastatic sites were associated with longer TNT in the first line, whereas age, mitotic index, mutational status, surgery of the primary tumor and LR procedure to metastatic sites were associated with longer OS. This real-life study advocates for early reference of metastatic GIST patients to expert centers to orchestrate the best access to future innovative clinical trials together with LR strategies and further improve GIST patients’ survival.

## 1. Introduction

Gastrointestinal stromal tumors (GIST) are rare mesenchymal tumors characterized by *KIT* or *PDGFRA* activating mutations [1]. Metastatic GIST are currently treated with oral KIT and PDGFR tyrosine-kinase inhibitors (TKI) such as imatinib, sunitinib and regorafenib [2,3,4]. Over two decades, significant changes in drug discovery and loco-regional (LR) procedures have impacted treatment strategies [5,6,7,8]. Recently, two new drugs have been approved in metastatic GIST, namely ripretinib after failure of at least three lines, including imatinib [9], and avapritinib in GIST with the *PDGFRA* exon 18 D842V mutation [10]. However, data about real-life multimodal treatment strategies and their outcomes for patients with metastatic GIST are limited [8].

We assessed evolution of real-life treatment strategies for patients with metastatic GIST treated in three French expert centers over 30 years, including access to clinical trials and LR procedures, and their impact on survival.

## 2. Materials and Methods

The national prospective database from the French Sarcoma Group (FSG) (https://conticabase.sarcomabcb.org/, accessed on 15 January 2020) was retrieved for:-Patients ≥ 18 years old, with a diagnosis of GIST;-with expert pathological review performed by members of RREPS (Réseau de Référence En Pathologie des Sarcomes);-treated in one of the three national coordinating centers from NETSARC (Institut Bergonié, Centre Léon Berard and Institut Gustave Roussy);-from 1990 to 2018;-who gave their informed consent to be included in the prospectively maintained FSG database;-and presented metastatic disease (metastatic at diagnostic or metastatic relapse).

The primary objective was to describe clinical and biological profiles as well as treatment modalities of patients with metastatic GIST in a real-life setting, including access to clinical trials and LR procedures in the metastatic setting. Secondary objectives were to assess (1) patients’ overall survival (OS) in the metastatic setting; (2) patients’ outcome in terms of time to next treatment (TNT) for each line of systemic treatment; (3) the impact of clinical trial inclusion and LR procedures on TNT and OS in the metastatic setting; (4) evolution of patients’ treatment modalities and OS among four therapeutics access periods: <2002 (pre-imatinib), 2002–2006 (pre-sunitinib), 2006–2014 (pre-regorafenib), post 2014; and (5) to determine clinical and molecular factors associated with TNT and OS in the metastatic setting, in univariate and multivariate analysis.

We collected available variables from the database, encompassing initial patients’ diagnostic characteristics (age at diagnosis, gender, previous history of specific disease); date of diagnosis, tumor characteristics at diagnostic (location, size, stage N, M, histology, mitotic index, Miettinen classification, mutational status); initial treatment modalities (surgery, margins, tumor spillage, neo/adjuvant systemic treatment, radiation therapy, other treatments (cryotherapy, radiofrequency, embolization), clinical trial inclusion); metastatic relapse/event characteristics (date of metastatic relapse, location of first metastasis, number of metastatic sites); treatment of metastatic event for each line (systemic therapy, date of start and stop, date of progression, inclusion in a clinical trial, LR treatment such as surgery, radiotherapy, cryotherapy, radiofrequency, embolization); and date of death.

Overall survival (OS) in the metastatic setting was defined as the interval between the diagnosis of metastatic disease (at diagnosis or relapse) and the time of death, from any cause. When death was not observed, OS was censored at the date of last patient contact. Time to next treatment (TNT) was defined as the time from the systemic treatment onset to the next treatment or death due to any cause, whichever came first. When neither death nor new systemic therapy was observed, TNT was censored at the date of last patient contact. A change in dosing of the same molecule made without changing the type of molecule was not considered a next treatment. This study was approved by institutional review boards.

Statistical analyses were performed using SAS 9.4 (SAS Institute Inc., Cary, NC, USA). Survival analyses were conducted using the Kaplan–Meier method and the log-rank test at a 0.05 two-sided significance level. The set of factors considered in univariate analysis was included in multivariate analysis. The remaining prognostic factors were selected using a backward stepwise selection method at the 0.05 level. The proportional hazards assumption and the functional form of a continuous variable were assessed with the proc phreg (proportionality_test and assess var options).

## 3. Results

Between 1990 and 2018, 1038 patients with a diagnosis of GIST made in one of the three participating centers were included in the national prospective database. Among them, 492 patients presented metastasis: 259 patients had metastatic disease at diagnosis and 233 patients developed a metastatic relapse. Patient characteristics are given in Table 1. Metastatic site was the liver for 181 (37%) patients, the peritoneum for 176 (36%) patients, and other monosite for 25 (5%) patients. One hundred and ten (22%) patients had multisite metastasis. The molecular distribution in the metastatic population was as follows: 58.3% of the patients had GIST with a mutation in *KIT* exon 11, 8.5% in *KIT* exon 9, 10.6% were *KIT/PDGFRA* wild type and 4.1% had a *PDGFRA* exon 18 D842V mutation, whereas 5.3% had another rare mutational status, including *KIT* exon 13, *KIT* exon 17, *PDGFRA* exon 12 and exon 14 mutations. 

Overall, 461 (94%) metastatic patients received systemic therapy in the metastatic setting, 244 (94%) patients with metastasis at diagnosis and 217 (93%) patients with metastatic relapse. Of those, 68 (31%) had received adjuvant/neoadjuvant systemic therapy for their primary event. The median number of therapy lines in the metastatic setting was 3 (range 0–15), and 171 (35%) patients received four or more lines. Patients received a large range of systemic therapies in addition to the three approved drugs (imatinib, sunitinib and regorafenib), including sorafenib, pazopanib, nilotinib, dasatinib, masitinib, dovitinib, cabozantinib as well as chemotherapy and anti-PD1/L1 (Figure 1). First line was imatinib in 384 (83%) patients. Second line was imatinib in 136 (45%) patients and sunitinib in 104 (34%) patients. Third line was sunitinib in 97 (40%) patients, imatinib in 58 (24%) patients, regorafenib in 19 (8%) patients, sorafenib and pazopanib in 13 (5%) patients each. 

A total of 269 patients (55%) participated in a clinical trial during the course of their metastatic disease, 177 (38%) in first line, 92 (30%) in second line, 62 (25%) in third line, 54 (32%) in fourth line and 39 (31%) in fifth line. Among the 259 patients with metastasis at diagnosis, 193 (74.5%) were offered surgery of their primary tumor. More than half (139 patients, 54%) underwent additional LR procedures to metastatic sites, including surgery in 125 (90%) patients, RF in 15 (11%) patients, radiotherapy in 10 (7%) patients and cryoablation in 2 (1.5%) patients. Twenty (14%) patients had multiple procedures. In the population of 233 patients with metachronous metastasis, almost half (110 patients, 47%) underwent additional LR procedures to metastatic sites, including surgery in 94 (85.5%) patients, RF in 20 (18%) patients, radiotherapy in 12 (11%) and cryoablation in 3 (3%) patients. Nineteen (17%) patients had multiple procedures.

The median follow-up was 108.9 months (95%CI [102; 120]). OS analysis was conducted in the whole study population of 492 patients with metastatic disease. The median OS in the metastatic setting was 83.4 months (95%CI [72.7; 97.9]) overall (Figure 2), 85.8 months (95%CI [68.1; 103.0]) for patients with metastatic disease at diagnosis and 82.9 months (95%CI [65.5; 101.0]) for patients with metachronous metastatic disease.

Time to next treatment (TNT) was analyzed in the population of 461 patients who received systemic therapy in the palliative setting. Overall, the median TNT was 26.7 months (95%CI [23.4; 32.3]) in the first line, 10.2 months (95%CI [8.6; 11.8]) in the second line, 6.7 months (95%CI [5.3; 8.5]) in the third line and 5.5 months (95%CI [4.3; 6.7]) in the fourth line, respectively. Figure 3 details TNT according to the main molecules used in each line. TNT in the first line (TNT1) was 33.8 months (95%CI [28.7; 38.2]) for imatinib and 8.2 months (95%CI [5.9; 13.9]) for other molecules (n = 461). Appendix A gives a description of baseline characteristics of patients who received imatinib (n = 384) or another molecule (n = 77) as first-line treatment, and shows no significant imbalance between both groups, notably in terms of molecular status and LR procedures. TNT in the second line (TNT2) was 12.7 months (95%CI [8.7; 15.5]) for imatinib and 11.2 months (95%CI [9.3; 14.9]) for sunitinib (n = 238). TNT in the third line (TNT3) was 10.8 months (95%CI [6.7; 17.9]) for imatinib, 5.4 months (95%CI [4.3; 8.5]) for sunitinib, and 9.5 months (95%CI [2.3; 18.6]) for regorafenib (n = 171). 

We then assessed OS and TNT in the first, second and third lines according to available molecular status. The median OS in the metastatic setting was 93.4 months (95% CI [78.4; 103.9]) for the *KIT* exon 11 mutation subgroup (n = 287), 68.4 months (95% CI [41.3; 78.3]) for the *KIT* exon 9 subgroup (n =42), not reached for the *PDGFRA* D842V subgroup (n = 20), and 59.2 months (95% CI [36.9; 107.8]) for the *KIT/PDGFRA* wild type subgroup (n = 52) (Appendix A).

TNT1 was 32.5 months (95% CI [26.0; 39.0]) in the *KIT* exon 11 subgroup, 22.7 months (95% CI [15.0; 31.6]) in the *KIT* exon 9 subgroup, 5.4 months (95% CI [3.9; NR]) in the *PDGFRA* D842V subgroup and 19.1 months (95% CI [5.5; 34.4]) in the *KIT/PDGFRA* wild type subgroup. TNT2 was 11.6 months (95% CI [9.6; 14.0]) in the *KIT* exon 11 subgroup, 8.9 months (95% CI [4.1; 14.9]) in the *KIT* exon 9 subgroup, 9.4 months (95% CI [2.8; 34.4]) in the *PDGFRA* D842V subgroup and 6.4 months (95% CI [3.5; 8.8]) in the *KIT/PDGFRA* wild type subgroup. TNT3 was 7.1 months (95% CI [5.3; 8.7]) in the *KIT* exon 11 subgroup, 5.1 months (95% CI [3.0; 13.6]) in the *KIT* exon 9 subgroup, 5.8 months (95% CI [2.5; NR]) in the *PDGFRA* D842V subgroup and 4.9 months (95% CI [2.1; 13.2]) in the *KIT/PDGFRA* wild type subgroup (Figure 4).

We finally assessed differences in OS and TNT among different periods of systemic therapy availability. Seventy patients were diagnosed with metastatic GIST in the first period (<2002, pre-imatinib approval), 98 patients in the second period (2002–2006, pre- sunitinib approval), 238 in the third period (2006–2014, pre-regorafenib approval), and 86 in the last period (diagnosed post 2014). Patients received a median of 3 lines in the first period [0–14], second period [0–15] and third period [0–12], and 1 line [0–6] in the last period. There was no statistical difference in OS in the metastatic setting between the four therapeutic periods (log rank *p* = 0.18) (Appendix A). The median TNT1 was 20.6 months (95%CI [12.5; 34.4]) in the first period, 32.5 months (95%CI [24.6; 38.5]) in the second period, 26.3 months (95%CI [20.7; 33.3]) in the third period, and 24.8 months (95%CI [16.6; NC]) in the last period. Median TNT2 was 13.7 months (95%CI [5.7; 22.1]) in the first period, 9.5 months (95%CI [7.1; 15.5]) in the second period, 9.9 months (95%CI [8.3; 11.6]) in the third period, and 12.1 months (95%CI [3.5; 18.8]) in the last period. The median TNT3 was 12.0 months (95%CI [5.1; 20.7]) in the first period, 6.3 months (95%CI [3.7; 10.3]) in the second period, 6.0 months (95%CI [4.9; 8.6]) in the third period and 5.7 months (95%CI [1.8; 7.7]) in the last period (Figure 5). 

In univariate analysis, age, gender, mitotic index, mutational status, surgery of primary tumor, margins of primary tumor resection, and LR procedure to metastatic sites were factors associated with OS in the metastatic setting (Appendix A). Factors that remained significant in multivariate analysis were age, mitotic index, mutational status, surgery of the primary tumor and LR procedure to metastatic sites (Table 2A). Surgery of the primary tumor remained an independent factor associated with OS in multivariate analysis in the cohort of 259 patients with synchronous metastasis, as well as size of the tumor, mitotic index and mutational status (Table 2B).

In univariate analysis, age, mitotic index, AFIP Miettinen classification, mutational status, surgery of primary tumor, margins of primary tumor resection, participation in a clinical trial at least once and in first line, and LR procedure to metastatic sites were factors associated with TNT in the first line (Appendix A). Factors that remained significant in multivariate analysis were age, AFIP Miettinen classification, mutational status, surgery of the primary tumor, participation in a clinical trial in the first line and the LR procedure to metastatic sites (Table 3).

## 4. Discussion

We report here real-life patterns of care of metastatic GIST patients treated in three large French expert centers over the three past decades. In this cohort of nearly 500 patients, the median OS in the metastatic setting was 83.4 months (95%CI [72.7; 97.9]), which is one of the highest reported to date. The median OS reported from first line imatinib in advanced and metastatic GIST patients ranges between 47 months in earlier studies and their follow-ups, 53 months in recent studies focusing on LR treatment in metastatic GIST, to 8.7 years in the study by Bauer et al. on 177 patients with R0/R1 surgery of their metastases [2,8,11,12,13]. 

The absence of a statistically significant difference in OS in the metastatic setting over the past three decades is quite unexpected. Two main reasons can explain these results. First is the high level of local procedures to metastasis performed in this study, and since early periods. ESMO guidelines validate considering local treatment of metastatic disease in two situations, that can be combined: in case of oligo-progression of a resistance clone, and in case of minimal residual disease after a good response, here also to minimize the risk of re-evolution of a resistant clone within the residual tumor, while continuing the ongoing TKI in both indications [14]. In our cohort of patients, three quarters of synchronous metastatic patients were offered surgery of their primary tumor, and half benefited from the locoregional procedure to their metastasis. Both were independent factors associated with OS in multivariate analysis. Of great interest, surgery of the primary tumor remained an independent prognostic factor for OS in the cohort of patients with synchronous metastatic disease, suggesting this approach is worthwhile in this population. Of note, only 37 patients (29,1%) from the recent study from Patterson et al. received palliative surgery, either for primary or metastasis [8]. Our results are in line with results from other expert network centers’ databases and advocate for surgery as an important tool in metastatic GIST patients when used by expert interdisciplinary teams [12,15]. Interestingly also, one third of patients with metachronous metastasis were offered locoregional procedures other than surgery, such as radiofrequency, cryoablation, or radiotherapy. There are very few data on impact of such procedures on GIST patients’ survival, mainly monocentric small series and case reports [6,8,16]. In our study, the LR procedure to metastatic sites was an independent factor associated with OS and TNT1.

The second reason that can explain our results is the high rate of inclusion in clinical trials. Indeed, more than half of our patients participated at least once in a clinical trial during the course of their metastatic disease, allowing access to innovative molecules earlier. The median OS of patients with metastatic GIST in the pre-imatinib approval period has been reported to be 18 months [2]. Our median TNT1 of 20 months in the first line in the pre-imatinib approval period illustrates the impact of giving earlier access to such pivotal molecules to patients in reference centers. Moreover, this access was constant over time, with about 30% of patients having access to clinical trials in up to the fifth line.

We chose to assess TNT as it is a more robust endpoint than PFS in retrospective designs [17,18], especially in GIST where there is no time off TKI, and when change in the molecule usually corresponds to disease progression. Progression in GIST is also difficult to assess, with CHOI and RECIST criteria having shown limits in this case [14,19,20,21]. Interestingly, the median TNT was 33.8 months in the first line with imatinib, 11.2 months in the second line with sunitinib and 9.5 months in the third line with regorafenib. These results are higher than PFS reported with the same molecules in the respective lines in pivotal clinical trials. Median PFS was indeed 20 months for imatinib in the first line, 6,8 months for sunitinib in the second line, and 4,8 months for regorafenib in the third line in the three pivotal clinical trials [2,3,4]. Of note, the median TNT in the fourth line was 5.5 months (95%CI [4.3; 6.7]) with various TKI in our study, which is in the range of the median PFS of 6.3 months [3.2; 8.2] reported with ripretinib in the INVICTUS trial in a non-molecularly selected population of patients with GIST after at least 3 lines [9]. Together with use of the LR procedure, these results can also be explained by improved toxicity management in time and dosing optimization in our three expert centers.

We assess OS and TNT in different lines according to molecular subtypes. Our data corroborate existing evidence on the major impact of mutational status on tumor evolution and response to available TKI [22,23,24,25]. Patients with the wild type of GIST had the worst prognosis, with an OS of only 59.2 months and TNT in the first, second and third lines of 19.1 months, 6.4 months and 4.9 months, respectively, despite important use of LR procedures. On the other hand, patients with a metastatic GIST bearing a PDGFRa D842V mutation had the best survival, with a median OS not reached at the time of data cutoff, but a disease refractory to existing TKI, illustrated by TNT in the first, second and third lines of 5.4 months, 9.4 months and 5.8 months, respectively. As awaited, patients with KIT exon 11 mutation had the best outcome, with a median OS of nearly eight years in the metastatic setting and TNT in the first and second lines of almost three and one year, respectively.

Our study used real world data (RWD) originating from a national prospective registry. RWD have been increasingly integrated into decision-making in oncology in the latest years. RWD analyses can offer valuable insights in situations where randomized controlled trials (RCTs) are not feasible, especially in rare tumors. In the other hand, when the outcomes observed in the RWD studies and in the RCT approximate each other, RWD reinforce evidence for RCTs results and can help extend inferences from RCTs to other patient populations or to longer follow-up schedules [26,27]. However, as a valuable alternative, observational RWD studies are also associated with multiple biases, such as prescription bias or bias of immortality and one should be cautious about adopting new therapies based on such evidence alone.

As awaited, the median TNT in the first line reached its highest value from the post imatinib registration period, but interestingly, the median TNT in the second line and further did not change significantly over the next decades, despite registration of sunitinib and regorafenib, and use of numerous other TKI such as sorafenib, dasatinib, pazopanib, cabozantinib, lenvatinib, etc. TNT also seemed to converge in advanced lines among main molecular subgroups. This suggests a plateau obtained in the global efficacy of the drugs used in GIST patients without taking into account secondary mutation occurrence as well as genomic instability and other pathway involvement [28,29]. These data underline the urgent need for a new generation of targeted drugs to specific secondary mutations and for designing clinical trials dedicated to specific molecular subtypes in GIST. Toward this path, Jones et al. reported a median PFS of 34 months (22.9—not reached) with avapritinib in pretreated patients with metastatic GIST bearing the specific PDGFRa D842V exon 18 mutation [10]. This trial with unprecedented benefits demonstrates the feasibility of successfully completing dedicated molecular trials in super rare tumors and the magnitude of patient survival improvement we can obtain. In the same vein, we are eager for the INSIGHT trial to start, assessing ripretinib versus sunitinib in the second line, dedicated to a specific population of patients with GIST bearing a mutation in KIT exon 11 associated with a mutation in KIT exon 17 (NCT05734105). The limit in such innovative designs is the diversity of molecular profiles of GIST leading to very small molecular subgroups of patients and the impossibility to perform randomized comparative phase 3 studies. This stresses the importance to refer patients with GIST to expert centers since the diagnosis of metastatic disease, to make sure they can have the best access to cutting edge molecular diagnostic tools together with innovative drugs and LR procedures to improve their survival. 

This study has limits. TNT can be longer than PFS when the physician does not change therapy despite progressing disease. However, this endpoint also embodies real life pragmatic strategies, and has a good correlation with OS [30], that was also assessed in this study with a robust follow up.

## 5. Conclusions

To conclude, this real-life study gives a comprehensive overview of multimodal therapeutic strategies for patients with metastatic GIST in three French expert centers over three decades. The median OS in the metastatic setting approached seven years and the median TNT in the first line exceeded two years. More than half of the patients participated in a clinical trial during the course of their metastatic disease and half underwent additional LR procedures on metastatic sites. Surgery of the primary tumor and LR procedures to metastatic sites were independent factors associated with prolonged OS and longer TNT in the metastatic setting, underlying that an aggressive multimodal approach is worthwhile in this disease, when performed by expert teams. Early access to clinical trials from the first line was also an independent factor associated with improved TNT. 

Importantly, we show that TNT and OS in metastatic GIST have not significantly evolved since the discovery of imatinib, despite access to numerous other TKI over time, and that a plateau has been reached with the non-molecularly selected strategy.

This real-life study advocates for early referral of GIST patients to expert centers to orchestrate the best access to future innovative clinical trials together with locoregional strategies and to further improve GIST patients’ survival.

## Figures and Tables

**Figure 1 cancers-15-04306-f001:**
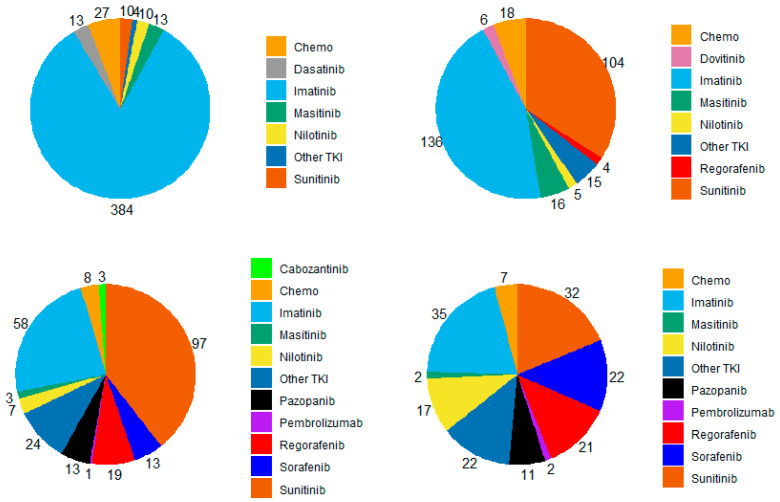
Repartition of systemic therapies used for each line of treatment in patients with metastatic GIST treated in three French expert centers over 30 years (n = 461). TKI: tyrosine kinase inhibitor; chemo: chemotherapy. (**Upper left**): 1st line, (**Upper right**): second line, (**Bottom left**): third line, (**Bottom right**): fourth line.

**Figure 2 cancers-15-04306-f002:**
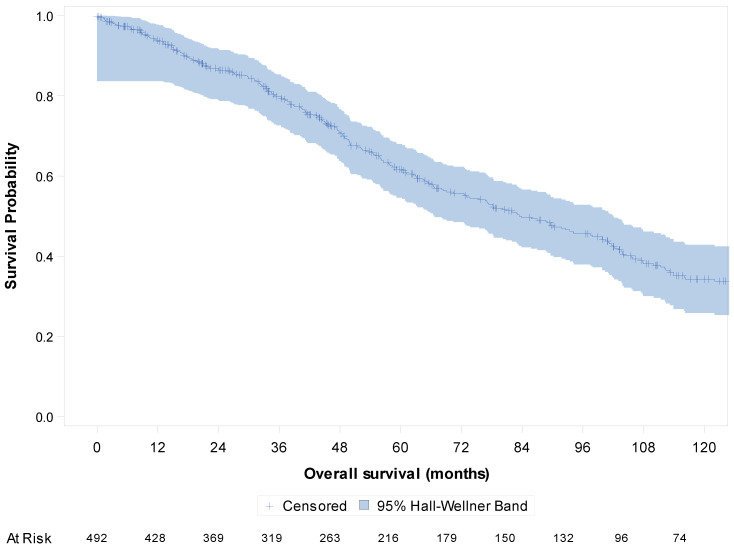
Kaplan–Meier curve of overall survival in the metastatic setting of the whole population (N = 492).

**Figure 3 cancers-15-04306-f003:**
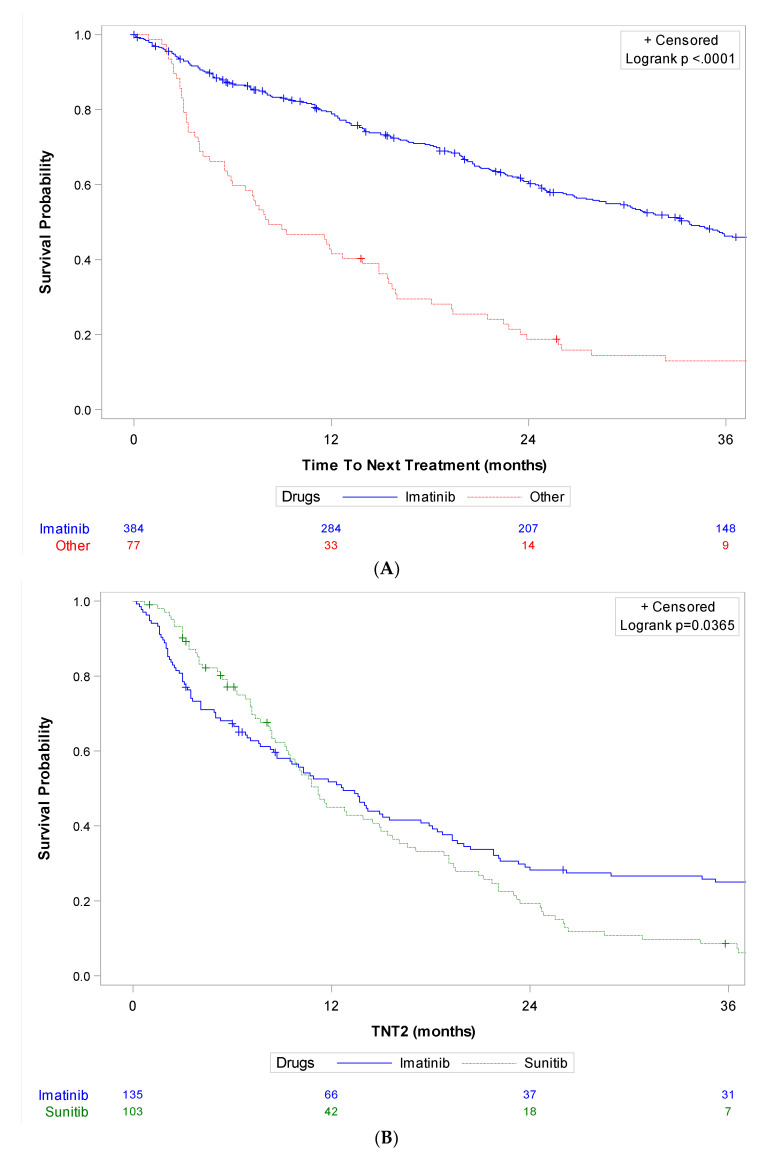
Kaplan–Meier curves of time to next treatment for each line of systemic therapy according to main molecules used in patients with metastatic GIST treated in three French expert centers over 30 years (n = 461). (**A**) Time to next treatment in the first line (TNT1) for imatinib vs. other molecules (n = 461), (**B**) Time to next treatment in the second line (TNT2) for imatinib and sunitinib (n = 238), (**C**) Time to next treatment in the third line (TNT3) for imatinib, sunitinib, and regorafenib (n = 171). Log-rank tests conducted for information purposes only, given the non-proportionality of risks.

**Figure 4 cancers-15-04306-f004:**
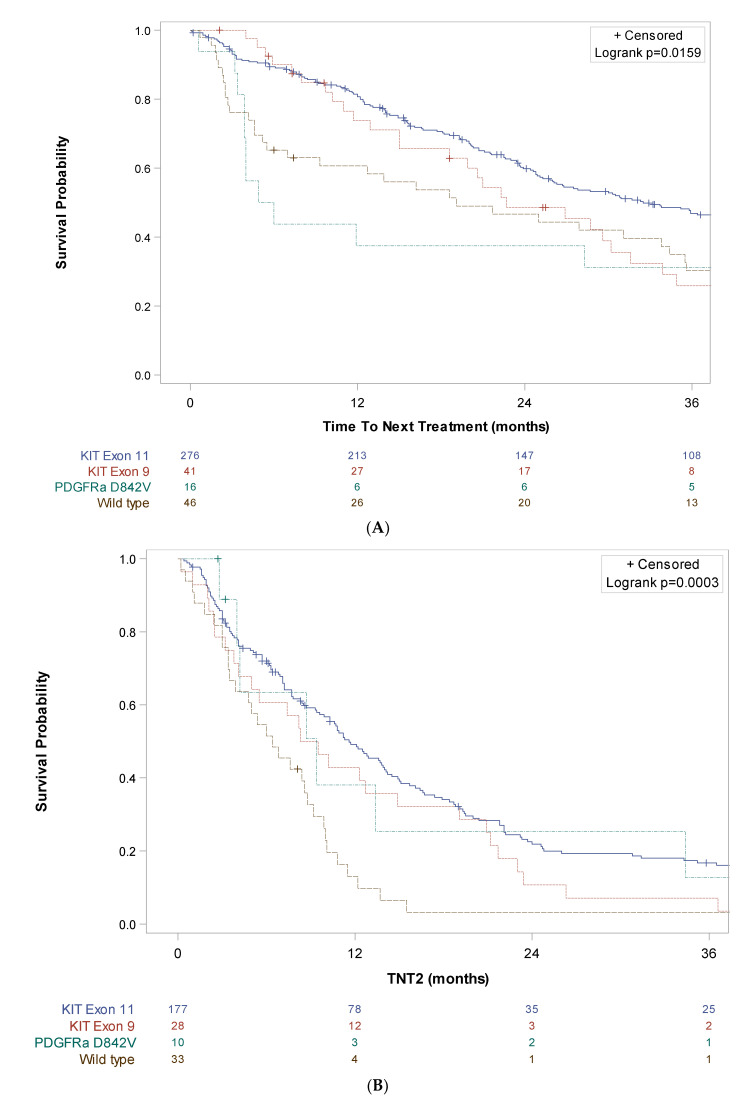
Kaplan–Meier curves of time to next treatment for each line of systemic therapy according to main molecular subgroup in patients with metastatic GIST treated in three French expert centers over 30 years (N = 401). (**A**) Time to next treatment in the first line (TNT1, A), in the second line (TNT2, (**B**)) and the third line (TNT3, (**C**)) in the KIT exon 11 (n = 287), KIT exon 9 (n = 42), PDGFRa D842V (n = 20) and wild type (n = 52) subgroups, respectively. Log-rank tests conducted for information purposes only, given the non-proportionality of risks.

**Figure 5 cancers-15-04306-f005:**
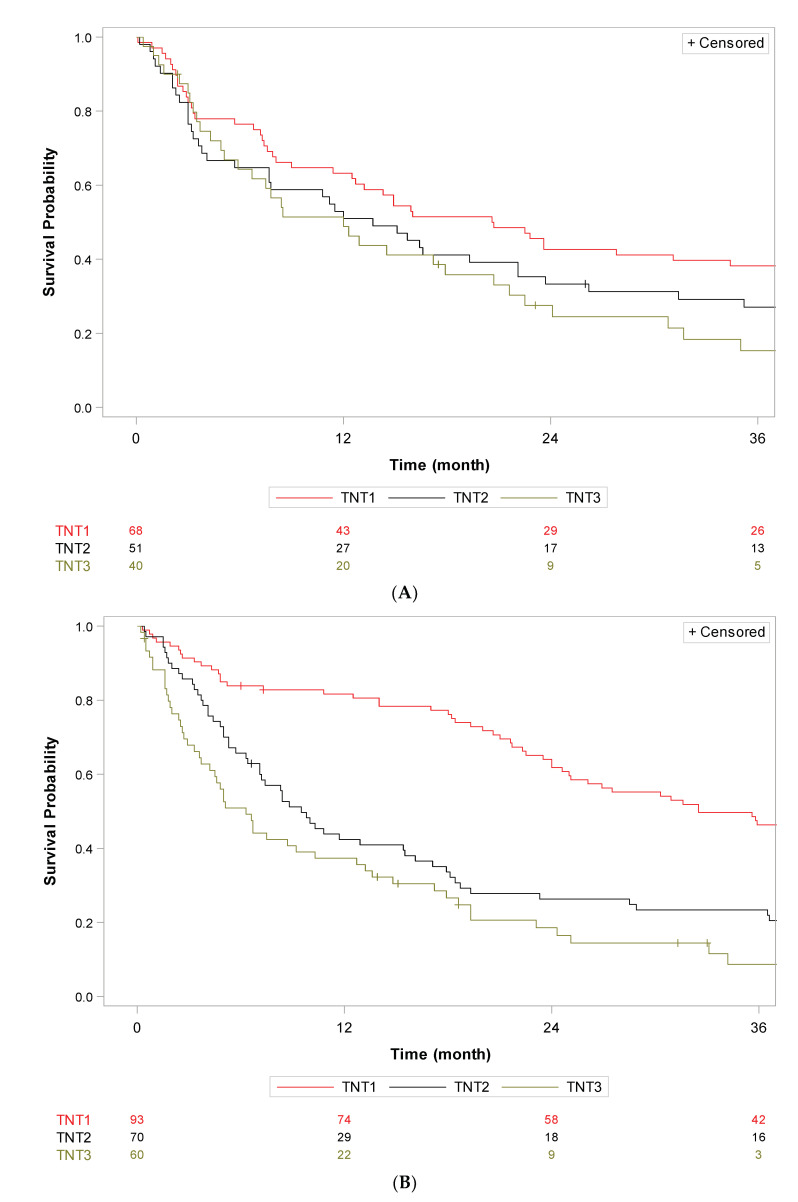
Kaplan–Meier curves of time to next treatment for each line of systemic therapy according to therapeutic periods with different access to TKI in patients with metastatic GIST treated in three French expert centers over 30 years (n = 461). TNT1, TNT2 and TNT3 for patients diagnosed (**A**) in the pre-imatinib era (<2002, n = 68), (**B**) in the pre- sunitinib era (2002–2006, n = 93), (**C**) in the pre-regorafenib era (2006–2014, n = 223), and (**D**) post 2014 (n = 77).

**Table 1 cancers-15-04306-t001:** Description of patients’ characteristics in study populations.

	All Patients (n = 1038)		Study Patients (n = 492)	
	n	%	n	%
**Sex**	547	52.7	292	59.3
Male				
Female	491	47.3	200	40.7
**Median age at diagnosis, years (min-max)**	61	(19–93)	59	(19–93)
**Significant previous history**				
No	850	81.9	424	86.2
Previous cancer	106	10.2	36	7.3
NF1	26	2.5	8	1.6
Other	21	2	7	1.4
Unknown	19	1.8	6	1.2
**Tumor site**				
Stomach	527	50.8	204	41.5
Small intestine	327	31.5	192	39.0
Duodenum	60	5.8	26	5.3
Rectum	55	5.3	24	4.9
Peritoneum	30	2.9	19	3.9
Colon	24	2.3	17	3.5
Esophagus	15	1.4	10	2.0
**Median tumor size, mm (min-max)**	80	(3–450)	100	(18–400)
**Median mitotic index/50HPF (min–max)**	5	(0–350)	10	(0–350)
**Miettinen AFIP scoring**				
High risk	501	48.3	340	69.1
Intermediate risk	186	17.9	69	14.0
Low risk	137	13.2	18	3.7
Very low risk	124	11.9	5	1.0
NA	90	8.7	60	12.2
**Mutational status**				
KIT Exon 11	543	52.3	287	58.3
KIT Exon 9	75	7.2	42	8.5
PDGFRa Exon 18 D842V	61	5.9	20	4.1
Wild type	115	11.1	52	10.6
Other *	77	7.4	26	5.3
NA	167	16.1	65	13.2
**Metastasis at diagnosis**				
No	774	74.6	**233**	**47.4**
Yes **	260	25.0	**259**	**52.6**
NA	4	0.4		
**Status At Expert Center Referral**				
First event	871	83.9	349	70.9
Metastatic Relapse/Progression	155	14.9	143	29.1
NA	12	1.2		

N.A.: Not Available, HPF: High PowerField. * In the metastatic cohort: KIT exon 13 (23%) and/or exon 17 (15%), or PDGFRa Exon 18 (42%), Exon 12 (19%), or Exon 14 (3%). ** One patient metastatic at diagnosis was excluded from the study because he was lost to follow up shortly after diagnosis.

**Table 2 cancers-15-04306-t002:** Prognostic factors associated with OS in multivariate analysis. (A) in the whole cohort of metastatic patients (n = 492); (B) in the cohort of patients with synchronous metastasis (n = 259).

A				
Clinical and Molecular Factors	*p*-Value (khi-2)	Hazard Ratio	HR Lower Conf. Limit	HR Upper Conf. Limit
**Age at diagnosis**				
Over 60 yo vs. under 60 yo	0.022	1.38	1.05	1.81
**Mitotic Index**	<0.001	1.01	1.00	1.01
**Mutational Status**	0.019			
PDGFRa D842V vs. KIT Exon 11	0.597	1.25	0.54	2.89
KIT Exon 9 vs. KIT Exon 11	0.011	1.80	1.14	2.85
Other vs. KIT Exon 11	0.507	1.26	0.64	2.50
Wild type vs. KIT Exon 11	0.002	1.90	1.26	2.86
NA vs. KIT Exon 11	0.910	1.03	0.66	1.60
**Surgery of primary tumor**				
Yes vs. No	0.001	0.48	0.31	0.76
**LR treatment of metastatic sites**				
Yes vs. No	<0.001	0.60	0.46	0.79
**B**				
**Clinical and molecular factors**	***p*-Value (chi-2)**	**Hazard Ratio**	**HR Lower Conf. Limit**	**HR Upper Conf. Limit**
**Tumor size**	0.004	1.00	1.00	1.01
**Mitotic Index**	0.015	1.01	1.00	1.01
**Mutational Status**	0.010			
PDGFRa D842V vs. KIT Exon 11	0.075	0.16	0.02	1.21
KIT Exon 9 vs. KIT Exon 11	<0.001	2.76	1.57	4.86
Other vs. KIT Exon 11	0.951	0.96	0.23	3.95
Wild type vs. KIT Exon 11	0.003	2.77	1.43	5.35
NA vs. KIT Exon 11	0.732	1.12	0.59	2.14
**Surgery of primary tumor**				
Yes vs. No	<0.001	0.34	0.21	0.57

**Table 3 cancers-15-04306-t003:** Prognostic factors associated with TNT in first line in multivariate analysis (n = 461).

Clinical and Molecular Factors	*p*-Value (khi-2)	Hazard Ratio	HR Lower Conf. Limit	HR Upper Conf. Limit
**Age at diagnosis**				
Over 60 yo vs. under 60 yo	0.037	1.28	1.01	1.61
**AFIP Miettinen classification**	0.006			
Intermediate risk vs. High risk	0.115	0.76	0.54	1.07
Low/very low risk vs. High risk	<0.001	0.30	0.15	0.61
NA/Missing vs. High risk	0.971			
**Mutational Status**	<0.001			
PDGFRa D842V vs. KIT Exon 11	0.004	2.25	1.29	3.93
KIT Exon 9 vs. KIT Exon 11	0.018	1.61	1.08	2.39
Other vs. KIT Exon 11	0.305	1.33	0.77	2.31
Wild type vs. KIT Exon 11	<0.001	1.95	1.36	2.81
NA vs. KIT Exon 11	0.008	1.70	1.15	2.50
**Surgery of primary tumor**				
Yes vs. No	0.005	0.56	0.38	0.84
**Participated in a clin. trial in 1st line**				
Yes vs. No	0.008	0.73	0.58	0.92
**LR treatment of metastatic sites**				
Yes vs. No	0.015	0.75	0.59	0.95

## Data Availability

Originla data are available on request to the corresponding author.

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
