# Peer review of "Evolution of Patterns of Care and Outcomes in the Real-Life Setting for Patients with Metastatic GIST Treated in Three French Expert Centers over Three Decades"

_cancers, 2023, doi:10.3390/cancers15174306_

Round 1

Reviewer 1 Report

This article on the therapy of patients with metastatic GIST treated in three expert centers in France over three decades maps the treatment reality of 492 patients with metastatic GIST. Contrary to the PFS mapped in trials, analyses are performed on time to next treatment (TNT), which again is more in line with the treatment reality. Of particular interest is the importance of local treatments, especially of the primary tumor, even in the metastatic stage, which may also explain the lack of difference in overall survival between the different treatment decades. In light of the randomized trials of second- and third-line therapies for GIST, the present analysis with a high case number and long survival times is of special interest.

Author Response

Thank you for this review and comments

Reviewer 2 Report

This is paper is fully documented, interpreted accounts of significant findings of original research and it provides new insights or interpretation of a subject through thorough and systematic evaluation of available evidence.  The paper is clearly written and well organized. The  introduction and the background are reasonable given the promise of the paper. The figures and tables are comprehensive and helpful

The paper generates the following kinds of data

1.      The clinical and biological profiles as well as treatment modalities of patients with metastatic GIST in a real-life setting, including access to clinical 40 trials and LR procedures in the metastatic setting.

2.      The assessement of patients outcome in terms of time to next treatment (TNT) for each line of systemic treatment, of patients overall survival (OS), then , the evolution of evolution of patients treatment modalities and OS according to treatment access : <   the impact of clinical trials and LR procedures on TNT and OS in the metastatic setting.

In conclusion the authors pointout that Age, AFIP Miettinen classification, mutational status, surgery of the primary tumor, participation in a clinical trial in first line and LR procedure to metastatic sites are associated with longer TNT in first line whereas age, mitotic index, mutational status, surgery of the primary tumor and LR procedure to metastatic sites are associated with longer OS in multivariate analysis. This real-life study advocates for early reference of metastatic GIST patients to expert centers to orchestrate best access to future innovative clinical trials together with LR strategies and further improve GIST patients survival.

Author Response

Thank you for this review and comments

Reviewer 3 Report

The article is well described, structured, but insufficiently discussed, taking into account the importance of biological and genetic factors. It is worth noting the importance of applying the results obtained. However, they are practically not described in the conclusion

The article is well described, structured, but insufficiently discussed, taking into account the importance of biological and genetic factors. It is worth noting the importance of applying the results obtained. However, they are practically not described in the conclusion

Author Response

Thank you for your review and comments. The article has been modified accordingly, including :

  • discussion on biological factors,
  • and enrichment of the conclusion with the results obtained.